# A new approach to the crystal habit retrieval from far infrared spectral radiance measurements

Gianluca Di Natale[1], Marco Ridolfi[1], and Luca Palchetti[1]

[1]National Institute of Optics, Via Madonna del Piano 10, Sesto Fiorentino, Firenze, Italy

**Correspondence:** Gianluca Di Natale (gianluca.dinatale@ino.cnr.it)

**Abstract.** To generate reliable climate predictions, global models need accurate estimates of all the energy fluxes contributing to the Earth Radiation Budget (ERB). Clouds in general, and more specifically ice clouds play a key role in the determination of the ERB, as they may exert either a feedback or a forcing action, depending on their optical and microphysical properties and physical state (solid/liquid). To date, accurate statistics and climatologies of cloud parameters are not available. Specifically, the ice cloud composition in terms of ice crystal shape (or habit) is one of the parameters with the largest uncertainty.

The Far-infrared Outgoing Radiation Understanding and Monitoring (FORUM) experiment, foreseen to be the $9^{th}$ Earth Explorer mission of the European Space Agency, will measure, for the first time spectrally resolved from space, the entire upwelling spectrum emitted by the Earth, from 100 to 1600 cm$^{-1}$. The far-infrared portion of the Earth spectrum, especially from 200 to 600 cm$^{-1}$, is very sensitive to cloud ice crystal shapes, thus FORUM measurements could also represent an opportunity to study the ice cloud composition in terms of ice crystal habit mixtures.

To investigate this possibility, we developed an accurate and advanced scheme allowing to model ice cloud optical properties also in cases of clouds composed of mixed ice crystal habits. This feature is in fact necessary, because also in situ measurements acquired over the years point out that the shape of ice cloud crystals varies depending on the crystal size range. In our model, the resulting cloud optical properties are also determined by the input habit fractions. Thus, the retrieval of these fractions from spectral radiance measurements can be attempted. Using 375 different cloudy scenarios, we assess the performance of our retrieval scheme in the determination of crystal habit mixtures starting from FORUM simulated measurements. The most relevant error components affecting the retrieved cloud parameters are not very large and are of random nature, thus FORUM measurements will allow to set up an accurate climatology of cloud parameters.

To provide an example of the benefit that one could get out of the habit mixture retrievals, we also show the improved accuracy of the thermal outgoing fluxes calculations as compared to using assumed mixtures.

## 1 Introduction

Several studies (Kiehl and Trenberth, 1997; Solomon, 2007) highlight the important contribution of clouds in the determination of Earth Radiation Budget (ERB). Clouds play a twofold role: if on one hand they contribute to cool the Earth system by reflecting back to space the incoming shortwave solar radiation, on the other hand they warm up the system by absorbing the longwave radiation emitted by the Earth's surface. The net radiative effect of the cloud depends on its type (Hartmann et al.,

1992), optical and micro-physical properties (Baran, 2009) and thermodynamic phase (Matus and L'Ecuyer, 2017). Depending on temperature and on supersaturation conditions (Bailey and Hallett, 2009), ice cirrus clouds may be composed of myriads of different crystal shapes (Baran, 2009), each shape being characterized by its own radiative properties (Yang et al., 2013). Furthermore, due to the variations of temperature and of water vapor concentration with height, clouds with large vertical extensions may be made of ice crystals with habits changing as a function of height. Specifically, while at the cloud top pristine crystals may be present, at the cloud bottom aggregates or other specific shapes may dominate (Baran, 2009).

Cloud radiative forcing (CRF) is defined as the difference between the radiation budget components for average cloud conditions and cloud-free conditions (Intrieri et al., 2002). The CRF particularly affects the surface radiation energy budget and is responsible for a fast and enhanced (as compared to mid-latitudes) warming of the polar regions (Stapf et al., 2020). For example, recent studies demonstrate that the total downwelling radiative flux in the internal regions of Antarctica changes from 50 to 220 W/m$^2$ (Bromwich et al., 2013; Di Natale et al., 2020) due to the forcing exerted by ice and supercooled water clouds, that are very frequent in polar regions (Cossich et al., 2021; Turner, 2005). This CRF leads to a net increase of the surface temperature (Stone et al., 1990) with respect to clear sky conditions depending on the cloud particle density, micro-physics and thermodynamic phase. For this reason it is important to quantify CRF through remote sensing measurements (Lolli et al., 2017; Campbell et al., 2021; Lewis et al., 2020).

Beside influencing the temperature of the polar ice sheets, as shown in Cess et al. (2001) and in Zhang et al. (2020), the CRF also impacts the temperature of the ocean's surface as the CRF changes its value also in connection with regional climatic events (like El Niño).

Despite the importance of the global energy budget for the study of the climate system, as pointed out by Wild (2012) and Stephens et al. (2012), the uncertainties in the various components of this budget are still large. For example, (Wild, 2020) shows that the net radiative flux absorbed by the Earth is estimated in the range 0.6÷1.2 W/m$^2$.A relevant fraction of this error is due to the uncertainty in the statistics of the ice cloud radiative properties. In fact, both the Far InfraRed (FIR) part of the outgoing longwave radiation (OLR) (Baran, 2007; Di Natale and Palchetti, 2022) and the reflectance in the visible/near infrared range (Cooper et al., 2006) are very sensitive to cloud particle size and ice crystal habits. Therefore, any uncertainty in these parameters directly maps onto the estimates of the radiative fluxes both at the top of the atmosphere and at the surface (Rossow et al., 1995).

A reliable statistics and an accurate parameterisation of ice clouds, accounting also for the crystal size and shape, would allow to better quantify the clouds effect on climate and the related climate feedbacks (McFarlane et al., 2005). As an example, Lubin et al. (1998) show that a better characterisation of the Antarctic ice cloud properties could improve the performance of the climate prediction models on the global scale.

The statistics of ice cloud properties are still affected by large uncertainties mainly due to the paucity of spectrally resolved measurements extending to the FIR region. The few FIR spectral measurements available to date are acquired either from ground (Di Natale et al., 2017; Garrett and Zhao, 2013; Maesh et al., 2001a; Di Natale et al., 2021; Rowe et al., 2019) or from aircraft / balloon (Bellisario et al., 2017; Costa et al., 2017; Bianchini et al., 2008) platforms, thus a global coverage is missing.

To date it is still a common practice to assume fixed ice crystal habit distributions in the retrieval of cloud properties from satellite, aircraft or ground-based radiances. However, the shape of ice crystals modulates the spectral radiance and a wrong shape assumption introduces biases in the simulated spectral radiances that, in the FIR region, may amount to 3–4 mW/(m$^2$ sr cm$^{-1}$) (Di Natale and Palchetti, 2022).

In some cases, like in Sourdeval et al. (2013); Iwabuchi et al. (2016); King et al. (2004); Li et al. (2005); Delanoë and Hogan (2010); Maesh et al. (2001b), the habit distribution is inferred from independent in-situ measurements. In other cases, like in Baum et al. (2005b) and Baum et al. (2007), size and habit distributions are extracted from band-averaged and spectrally resolved climatologies based on the measurements performed in several previous field campaigns (Baum et al., 2005a).

With the aim to better characterise ice habit distributions, Chepfer et al. (2002) developed and tested a couple of new techniques: the dual-satellite retrieval method and the lidar depolarisation classification method. In the first method, a given cloud area is observed by two satellites from different directions (McFarlane et al., 2005; Chepfer et al., 2002) or from a satellite with an instrument capable of measuring reflectance at different angles, such as the POLarization and Directionality of Earth Reflectance (POLDER) on board of the Advanced Earth Observing Satellite (ADEOS) and the Polarization and Anisotropy of Reflectances for Atmospheric Sciences coupled with Observations from a Lidar (PARASOL) (Chepfer et al., 2002; Cole et al., 2014) or also the Along Track Scanning Radiometer (ATSR) on board of the European Remote Sensing Satellite (ERS-2) (Baran et al., 1998). This method exploits the fact that the radiance ratio at the two view directions is driven by the behaviour of the phase function of the different habits. The second method is based on the fact that different habits produce different depolarisations on the backscattering lidar signal. For example, plates produce low depolarization in the same range of aspect ratio as spherical particles, so both shapes are equivalent in the classification scheme, instead, columnar particles typically produce large values (Chepfer et al., 2002; Sourdeval et al., 2013). Both these methods show some limits, in particular the dual-satellite method requires the alignment of two satellites and when using the reflectance, only the thickest clouds are processed (Cole et al., 2014). Furthermore, the POLDER instrument has already been dismissed and the lidar techniques can only approximately estimate which type of shape dominates in the cloud.

For these reasons, the uncertainties on ice crystal habits make the retrieval of ice cloud parameters a complicated and challenging problem (L'Ecuyer et al., 2006). Despite of that, the different sensitivity of the various spectral intervals, from the FIR to the Middle-InfraRed (MIR), to the crystal shapes can be exploited to discriminate the content of each habit in the cloud. This is similar to what is usually done for the determination of the ice/liquid fraction in mixed-phase clouds, thanks to the different spectral behaviour of ice and water refractive indices (Turner et al., 2003).

In Di Natale and Palchetti (2022), we presented a simplified inversion algorithm able to retrieve ice crystal habit fractions from nadir-looking spectral radiance measurements extending from the FIR to the MIR. In the present work, we present a more sophisticated scheme that overcomes the approximations used in Di Natale and Palchetti (2022), by exploiting a more elaborated set of lookup tables, specifically developed for this application. So far, the method is limited to single layer homogeneous clouds covering the whole instrument field of view.

The Far-infrared Outgoing Radiation for Understanding and Monitoring (FORUM) (Palchetti et al., 2020, 2016; Ridolfi et al., 2020) and The Polar Radiant Energy in the Far Infrared Experiment (PREFIRE) (L'Ecuyer et al., 2021) are two satel-

lite missions planned for launch in the near future. Both missions will provide, with different resolutions, spectral radiance measurements in the FIR and MIR regions of the terrestrial spectrum from space. These measurements will allow to check the accuracy and the self-consistency of the state-of-the-art cloud radiative models across the whole OLR spectral range. Potentially, from these measurements it shall also be possible to retrieve ice cloud habit distributions. In this paper, we test our new developed forward / retrieval method on FORUM synthetic measurements, and assess the performance and the characteristics of the cloud parameters that may be inferred. The method is still to be validated on the basis of real measurements.

The paper is organized as follows. In Section 2, we introduce the new forward / retrieval scheme to handle the optical properties of ice clouds consisting of a mixture of crystal habits. In the same Section, we also quantify the radiance errors implied by the approximations adopted in Di Natale and Palchetti (2022), and examine the sensitivity of the FORUM measurements to crystal habit fractions. Section 3 discusses the results of a self consistency test for the retrieval and presents the characteristics of the habit fractions retrieved from FORUM measurements simulated on the basis of the anticipated instrument specifications. In Section 4, we quantify the most relevant model error components that are expected to affect the retrieved cloud parameters and crystal habit fractions. In Section 5, we show the reduction of the error in the calculated OLR fluxes obtained when retrieving the ice crystal habit fractions. Finally, in Section 6, we draw the conclusions and analyse the future perspectives.

## 2    Modelling the ice cloud optical properties

### 2.1    Optical properties of a mixture of ice crystal habits

At each wavenumber $\nu$, the radiative transfer through atmospheric gases in the presence of a generic aerosol, like clouds, can be simulated by means of the aerosol bulk properties, such as the average monochromatic extinction, absorption and scattering efficiencies: $\langle Q_\mathrm{e} \rangle_\nu$, $\langle Q_\mathrm{a} \rangle_\nu$ and $\langle Q_\mathrm{s} \rangle_\nu$, respectively. From these quantities it is possible to derive the single scattering albedo $\langle \omega \rangle_\nu$ and the asymmetry factor $\langle g \rangle_\nu$ describing the behaviour of the scattering function. For each crystal habit $h$, the parameters $Q_{\mathrm{e},h\nu}(L)$, $Q_{\mathrm{a},h\nu}(L)$ and $g_{h\nu}(L)$ are available from specific databases (Yang et al., 2013; Bi and Yang, 2014, 2017), where they are tabulated versus wavenumber (in the range $100 \leq \nu \leq 50000$ cm$^{-1}$), and maximum crystal length $L$, for $2 \leq L \leq 10^4$ $\mu$m.

The average values of the extinction, absorption and scattering efficiencies, of the single scattering albedo and of the asymmetry factor can be calculated by integration, assuming a particle size distribution (PSD) $n(L, L_\mathrm{m})$, with the parameter $L_\mathrm{m}$ proportional to the mode (Yang et al., 2005) of the PSD. More specifically:

$$\langle Q_\mathrm{e} \rangle_\nu = \frac{\sum_{h=1}^{N} p_h \int_{L_\mathrm{min}}^{L_\mathrm{max}} A_h(L) Q_{\mathrm{e},h\nu}(L) n(L, L_\mathrm{m}) dL}{\sum_{h=1}^{N} p_h \int_{L_\mathrm{min}}^{L_\mathrm{max}} A_h(L) n(L, L_\mathrm{m}) dL} \tag{1}$$

$$\langle Q_\mathrm{a} \rangle_\nu = \frac{\sum_{h=1}^{N} p_h \int_{L_\mathrm{min}}^{L_\mathrm{max}} A_h(L) Q_{\mathrm{a},h\nu}(L) n(L, L_\mathrm{m}) dL}{\sum_{h=1}^{N} p_h \int_{L_\mathrm{min}}^{L_\mathrm{max}} A_h(L) n(L, L_\mathrm{m}) dL} \tag{2}$$

$$\langle \omega \rangle_\nu = 1 - \frac{\langle Q_{\mathrm{a}} \rangle_\nu}{\langle Q_{\mathrm{e}} \rangle_\nu} \tag{3}$$

$$\langle g \rangle_\nu = \frac{\sum_{h=1}^{N} p_h \int_{L_{\min}}^{L_{\max}} A_h(L) Q_{\mathrm{s},h\nu}(L) g_{h\nu}(L) n(L,L_{\mathrm{m}}) dL}{\sum_{h=1}^{N} p_h \int_{L_{\min}}^{L_{\max}} A_h(L) Q_{\mathrm{s},h\nu}(L) n(L,L_{\mathrm{m}}) dL} \tag{4}$$

where the scattering efficiency $Q_{\mathrm{s},h\nu}(L)$ is given by $Q_{\mathrm{s},h\nu}(L) = Q_{\mathrm{e},h\nu}(L) - Q_{\mathrm{a},h\nu}(L)$. The above expressions, are calculated for a mixture of crystal habits defined by the relative fractions $p_h$, with $h = 1, \cdots, N$. As $p_h$ are relative fractions, they must fulfill the normalization condition:

$$\sum_{h=1}^{N} p_h = 1. \tag{5}$$

$L_{\min}$ and $L_{\max}$ are the boundaries of $L$ used to generate the database of the single scattering crystal properties, thus, as mentioned above, they are equal to 2 and $10^4$ $\mu$m, respectively. $V_h$, $A_h$ and $Q_{s,h\nu}$ denote the volume, the projected area and the scattering efficiency of each crystal habit type.

As suggested by Platnick et al. (2017) and McFarlane and Marchand (2008), we assume the particle size distribution to be a gamma function:

$$n(L,L_{\mathrm{m}}) = N_o L^\mu e^{-(\mu+3)\frac{L}{L_{\mathrm{m}}}} \tag{6}$$

where $N_o$ is the intercept, $\mu$ the dispersion coefficient and the quantity $R_m = L_{\mathrm{m}}/(\mu + 3)$ denotes the modal length or the inverse of the slope of the PSD. We define the effective diameter of ice particles $D_e$ as (Yang et al., 2013):

$$D_{\mathrm{e}} = \frac{3}{2} \frac{\sum_{h=1}^{N} p_h \int_{L_{\min}}^{L_{\max}} V_h(L) n(L,L_{\mathrm{m}}) dL}{\sum_{h=1}^{N} p_h \int_{L_{\min}}^{L_{\max}} A_h(L) n(L,L_{\mathrm{m}}) dL} \tag{7}$$

As shown in Wyser and Yang (1998), using this expression for $D_{\mathrm{e}}$ makes the optical properties of ice crystals insensitive to the detailed shape of the size distribution. Using this expression and Eq. (1), the optical depth at wavenumber $\nu$ ($\mathrm{OD}_\nu$) can be calculated as (Yang et al., 2003):

$$\mathrm{OD}_\nu = \frac{3 \cdot \mathrm{IWP}}{D_{\mathrm{e}} \rho_{\mathrm{i}}} \frac{\langle Q_{\mathrm{e}} \rangle_\nu}{2} = \mathrm{OD} \frac{\langle Q_{\mathrm{e}} \rangle_\nu}{2} \tag{8}$$

where OD is the optical depth at visible wavelength, IWP is the Ice Water Path and $\rho_{\mathrm{i}} = 917$ kg m$^{-3}$ is the ice density. The properties given by expressions (1) - (4) and (7), (8) allow to fully describe the interaction of a cloud composed of ice crystals with different habits with the radiation, and to simulate the radiative transfer in the atmosphere.

Note that in the approach described in Di Natale and Palchetti (2022), the bulk optical efficiencies are obtained as a weighted sum of the individual habit optical efficiencies. The weights of that sum are the habit fractions. Furthermore, a single effective diameter is assumed for all habits. Apparently, that approach is an approximation because in the rigorous Eqs. (1) and (2), the contributions of the individual habits are clearly not separable. As explained in the next Section, in the new proposed approach the complexity of Eqs. (1) and (2) is tackled by tabulating separately each of the integrals appearing in the above equations.

## 2.2 Building new look up tables

The relative habit fractions $p_h$ are unknowns that, together with the size distribution parameter $L_{\mathrm{m}}$ and with the optical depth OD, we would like to retrieve from atmospheric spectral radiance observations. The retrieval is achieved by the iterative minimization of a cost function (CF) with the evaluation of expressions (1) - (8) at each iteration. From the computational point of view, the explicit evaluation at each iteration of all the integrals appearing in those expressions would be a very costly operation. For this reason, we decided to pre-compute and tabulate as a function of wavenumber $\nu$ and parameter $L_{\mathrm{m}}$ all the integrals appearing in expressions (1), (2), (4) and (7). More specifically, for each habit $h$ (hollow columns (HC), solid bullet rosettes (SBR), droxtals (DX) and plates (PL)), we tabulated the following integrals for $100 \le \nu \le 1600$ cm$^{-1}$ and $2 \le L_{\mathrm{m}} \le 10^4$ $\mu$m (step 2 $\mu$m):

$$\langle Q'_{e,h}\rangle_\nu = \int_{L_{\min}}^{L_{\max}} A_h(L)Q_{e,h\nu}(L)n(L,L_{\mathrm{m}})dL, \tag{9}$$

$$\langle Q'_{a,h}\rangle_\nu = \int_{L_{\min}}^{L_{\max}} A_h(L)Q_{a,h\nu}(L)n(L,L_{\mathrm{m}})dL, \tag{10}$$

$$\langle g'_h\rangle_\nu = \int_{L_{\min}}^{L_{\max}} A_h(L)Q_{s,h\nu}(L)g_{h\nu}(L)n(L,L_{\mathrm{m}})dL, \tag{11}$$

$$\langle V'_h\rangle = \int_{L_{\min}}^{L_{\max}} V_h(L)n(L,L_{\mathrm{m}})dL, \tag{12}$$

$$\langle A'_h\rangle = \int_{L_{\min}}^{L_{\max}} A_h(L)n(L,L_{\mathrm{m}})dL. \tag{13}$$

To compute the optical properties, we then linearly interpolate the tabulated quantities to the required values of $\nu$ and $L_{\mathrm{m}}$, and evaluate Eqs. (1), (2), (4) and (7) as:

$$\langle Q_e\rangle_\nu = \frac{\sum_{h=1}^N p_h\langle Q'_{e,h}\rangle_\nu}{\sum_{h=1}^N p_h\langle A'_h\rangle}, \tag{14}$$

$$\langle Q_a\rangle_\nu = \frac{\sum_{h=1}^N p_h\langle Q'_{a,h}\rangle_\nu}{\sum_{h=1}^N p_h\langle A'_h\rangle}, \tag{15}$$

$$\langle g \rangle_\nu = \frac{\sum_{h=1}^N p_h \langle g_h' \rangle_\nu}{\sum_{h=1}^N p_h \left( \langle Q_{e,h}' \rangle_\nu - \langle Q_{a,h}' \rangle_\nu \right)}, \tag{16}$$

$$D_e = \frac{3}{2} \frac{\sum_{h=1}^N p_h \langle V_h' \rangle}{\sum_{h=1}^N p_h \langle A_h' \rangle}. \tag{17}$$

## 2.3 Retrieval setup

The test retrievals presented in this work are carried out using the Simultaneous Atmospheric and Clouds Retrieval (SACR)
code developed at our premises (Di Natale et al., 2020). SACR is based on the optimal estimation approach of Rodgers (2000).
In this approach, the solution corresponds to the state $\mathbf{x}$ that minimizes the following cost function:

$$\chi^2(\mathbf{x}) = (\mathbf{y} - \mathbf{f}(\mathbf{x}))^T \mathbf{S}_{\mathrm{y}}^{-1} (\mathbf{y} - \mathbf{f}(\mathbf{x})) + (\mathbf{x} - \mathbf{x}_{\mathrm{a}})^T \mathbf{S}_{\mathrm{a}}^{-1} (\mathbf{x} - \mathbf{x}_{\mathrm{a}}), \tag{18}$$

where $\mathbf{y}$ is the $m$-dimensional measurement vector with associated error covariance matrix $\mathbf{S}_{\mathrm{y}}$, $\mathbf{f}(\mathbf{x})$ is a radiative transfer
model simulating the measurement $\mathbf{y}$ from the atmospheric, cloud and surface state $\mathbf{x}$. The vector $\mathbf{x}_{\mathrm{a}}$ is an estimate of $\mathbf{x}$, with
185 error covariance matrix $\mathbf{S}_{\mathrm{a}}$. In the absence of biases in the measurements and in the a priori, the expectation value of $\chi^2$ is
equal to the dimension $m$ of the measurement $\mathbf{y}$, thus we often refer to the normalized cost function $\chi_{\mathrm{N}}^2(\mathbf{x}) = \chi^2(\mathbf{x})/m$ with
expectation value equal to 1. The inversion algorithm finds the minimum of $\chi^2(\mathbf{x})$ iteratively, with the Levenberg-Marquardt
(LM) technique (Levenberg, 1944; Marquardt, 1963). The iterations are stopped when the variation of the cost function within
two subsequent iterations is less than $10^{-4}$.
In the application presented here, SACR uses a retrieval state vector given by:

$$\mathbf{x} = (\mathbf{p}, L_{\mathrm{m}}, \mathrm{OD}) \tag{19}$$

where $\mathbf{p} = (p_1, \cdots, p_N)$ is the vector of habit fractions, while the other symbols where already introduced in Sect. 2.1. Surface
spectral emissivity and temperature, as well as atmospheric profiles are assumed as known. To guarantee that the retrieved
habit fractions $p_h$ meet the constraint (5), we apply a change of variables to this section of the state vector, following the
195 same approach described in Di Natale and Palchetti (2022). The inversion algorithm operates on the $N - 1$ internal variables
$\mathbf{q} \to \mathbf{p}(\mathbf{q}) = (p_1(\mathbf{q}), p_2(\mathbf{q}), \cdots, p_{N-1}(\mathbf{q}))$ with $\mathbf{q} = (q_1, q_2, \cdots, q_{N-1})$. The components of the vector $\mathbf{q}$ are constrained so
that $q_k \in [0,1]$ for $k = 1, .., N - 1$. The fractions $p_h$ are then obtained from the following back-transformation (Di Natale and

Palchetti, 2022):

$$
\begin{cases}
p_1(\mathbf{q}) = q_1 \\
p_2(\mathbf{q}) = q_2(1 - p_1(\mathbf{q})) = q_2(1 - q_1) \\
p_3(\mathbf{q}) = q_3(1 - p_1(\mathbf{q}) - p_2(\mathbf{q})) = q_3(1 - q_1 - q_2(1 - q_1)) \\
\vdots \\
p_{N-1}(\mathbf{q}) = q_{N-1}(1 - p_1(\mathbf{q}) - \cdots - p_{N-2}(\mathbf{q})) = q_{N-1}(1 - \sum_{h=1}^{N-2} p_h(\mathbf{q})).
\end{cases}
\tag{20}
$$

The last coefficient $p_N$ is determined on the basis of the normalisation condition:

$$
p_N(\mathbf{q}) = 1 - \sum_{h=1}^{N-1} p_h(\mathbf{q})
\tag{21}
$$

As shown in Di Natale and Palchetti (2022), this transformation is invertible and ensures both the normalization condition (5) and that $p_k \in [0, 1]$ for $k = 1, .., N$. Using the formulas presented in Di Natale and Palchetti (2022), we evaluate the error covariance matrix $\mathbf{S}_x$, mapping the measurement noise error onto the solution of the retrieval state $\mathbf{x}$.

## 2.4 Radiance differences implied by the approximation of Di Natale and Palchetti (2022)

In Di Natale and Palchetti (2022), we illustrate a scheme that represents a first attempt to retrieve ice crystal habit fractions from spectral radiance observations. Instead of using the rigorous formulas presented in Sect. 2.1, in Di Natale and Palchetti (2022) the bulk cloud optical properties of the habit mixture are approximated with a linear combination of the optical properties relating to each individual crystal habit. In this Section, we estimate the radiance error implied by that approximation by comparison to the rigorous calculations proposed in this work.

As a case study, we use the simulated upwelling spectral radiances corresponding to a mid-latitude atmosphere extracted from the ECMWF-ReAnalysis 5 (ERA5) dataset of the European Centre for Medium-Range Weather Forecasts (ECMWF) for the day of 05/08/2019 at 12:00:00 UTC (Hersbach et al., 2020). In the selected scenario, the surface temperature is equal to 15 °C and we assume the surface as covered by grass, with spectral emissivity given by Huang et al. (2016). An ice cloud layer exists between 8 and 10 km above the ground. We assume the cloud as consisting of an homogeneous mixture of crystals with four different habits: hollow columns (HC), solid bullet rosettes (SBR), droxtals (DX) and plates (PL). As explained in King et al. (2004), on the basis of climatology (Cole et al., 2014; McFarlane and Marchand, 2008), four different habits are generally considered sufficient to describe a cloud. We perform two sets of simulations assuming, respectively, $L_m = 40$ $\mu$m ($D_e = 23$ $\mu$m) and $L_m = 200$ $\mu$m ($D_e = 100$ $\mu$m). In each of the two sets of simulations, OD spans the range from 0.001 to 150. In all the simulations, the ice crystal optical properties are extracted from the databases of Yang et al. (2013). Although this is not the most recent version of the ice crystal optical properties released by the Yang's team (see Bi and Yang 2014, 2017), we preferred to use this version for consistency and inter-comparability with the simplified approach of Di Natale and Palchetti (2022). This choice does not actually affect the reliability of the results presented here, which are based on retrievals from synthetic measurements. However, it is clear that when analysing real measurements a switch to the most

recent release of ice crystal optical properties will be necessary. To enable the comparison of the linear approximation error to the measurement error expected from the forthcoming FORUM measurements, the simulated upwelling spectral radiances are convolved with a *sinc* function with a Full Width at Half Maximum of 0.5 cm$^{-1}$, emulating the expected FORUM spectral response function (Ridolfi et al., 2020).

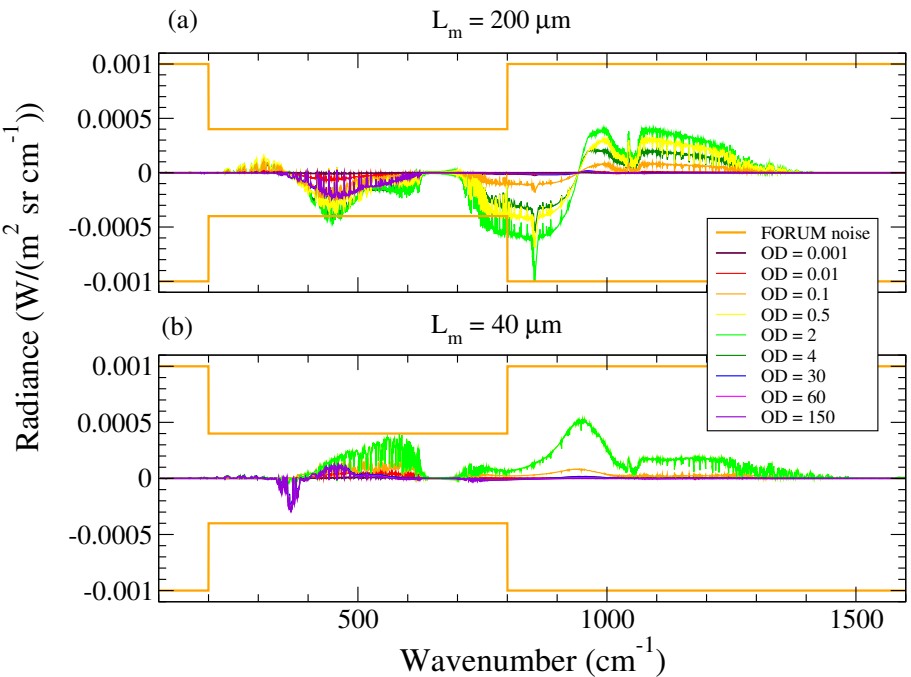

**Figure 1.** Differences between spectral radiances computed with the rigorous approach proposed in this work and with the approximation of Di Natale and Palchetti (2022). Panels (a) and (b) refer to simulations with $L_m$ equal to 200 and 40 $\mu$m, respectively. The coloured lines refer the different OD values, as indicated in the plot's key. FORUM measurement noise is also shown for reference (orange line).

Figure 1 shows the differences between the spectral radiances computed with the rigorous approach proposed in this work
and with the approximation of Di Natale and Palchetti (2022). Panels (a) and (b) refer to the simulations with $L_m$ equal to 200 and 40 $\mu$m, respectively. For reference, Fig. 1 also shows the noise level expected for the radiance measurements of the FORUM mission (in orange, Ridolfi et al. 2020). The coloured lines refer to various OD values, as indicated in the plot's key. Explaining the behavior of these differences on the basis of the radiative transfer equation and of the expressions used to compute the various contributing elements (see Di Natale et al. 2020) is a very complicated issue. Despite of this complication,
we see that the asymptotic behavior of the differences, for OD→0 and for OD→ ∞ is reasonable. Specifically, for OD→0, i.e. for ice amounts getting closer and closer to zero, the cloud effect on the upwelling spectral radiance must get closer and closer to zero, thus the two compared methods should provide the same result. This is confirmed by the lines of Fig. 1 corresponding to the smallest OD values. Conversely, in the presence of a very opaque cloud (OD>>1), the radiance should depend uniquely on the absorption and scattering processes occurring at the cloud top. Therefore, we expect the differences between the radiance

predicted by the two methods to approach a wavenumber-dependent asymptotic value that does not actually change for any further increase of cloud OD. Looking at the lines of Fig. 1 that correspond to OD$\geq$ 30, we see that they almost overlap, thus confirming the expected behavior. Finally, note that the differences between the two methods increase for increasing OD, reach a maximum amplitude for OD$\approx$ 2, then decrease to their asymptotic value for OD$>>$1.

We clearly see that in case of optically thin clouds, the radiance error caused by the linear combination approximation is much smaller than the FORUM noise. However, for optically thicker clouds this error becomes comparable to the amplitude of the FORUM noise. Considering that this is a *model* error, thus spectrally correlated, this may imply a systematic error component on the parameters retrieved from the measurement, with amplitude even larger than that of the error due to the measurement noise.

## 2.5 Sensitivity of the radiance to changes in crystal habit mixture

In this Section, we check the sensitivity of the upwelling spectral radiances to ice habit mixture in the cloud, assuming the same atmospheric and surface states described in Sect. 2.4. Also in this scenario, the ice cloud layer is located between 8 and 10 km. We consider two cloud cases, both with OD=1, differing for the value of $L_{\mathrm{m}}$ that is equal to 40 $\mu$m in the first case and to 200 $\mu$m in the second case. The reference radiance simulation includes an ice cloud with a uniform mixture ($p_1 = p_2 = p_3 = p_4 = 0.25$) of four crystal habits: HC, SBR, DX and PL. Then, we carry out four *perturbed* simulations each corresponding to a +0.6 increase of the fraction of a given habit and a -0.2 decrease of the other three habits, so that the normalization $p_1 + p_2 + p_3 + p_4 = 1$ is preserved. As in Sect. 2.4, all the radiances are convolved with the spectral response function expected for FORUM measurements.

Figure 2 shows the differences between radiances obtained with perturbed and reference (uniform) habit mixtures. As indicated in the plot's key, the curve colors refer to the specific habit whose fraction has been increased for the generation of the perturbed simulation. Panel (a) of the figure refers to the cloud with $L_{\mathrm{m}} = 40$ $\mu$m, while panel (b) refers to the cloud with $L_{\mathrm{m}} = 200$ $\mu$m. We see that both the FIR interval between 300 and 600 cm$^{-1}$ and the MIR interval from 750 to 1350 cm$^{-1}$, are very sensitive to all the habit mixture variations considered, with radiance changes exceeding by far the FORUM expected measurement noise. Note also that the spectral shapes of the radiance differences obtained by perturbing the various habits are different from each other, with differences exceeding the FORUM noise. This means that FORUM measurements, at least in the examined scenario, contain sufficient information also to discriminate between the different habit fractions.

## 3 Test retrievals from FORUM simulated measurements

### 3.1 Observational scenarios

The atmospheric and surface scenarios, we selected to test the retrieval of ice cloud habits, are based on the ERA5 dataset. Among the data included therein, we use the surface type and temperature, and the vertical profiles of temperature, humidity, ozone and ice water content (IWC(z)). We exploit the profile of IWC(z) only to define the geometrical extension and the

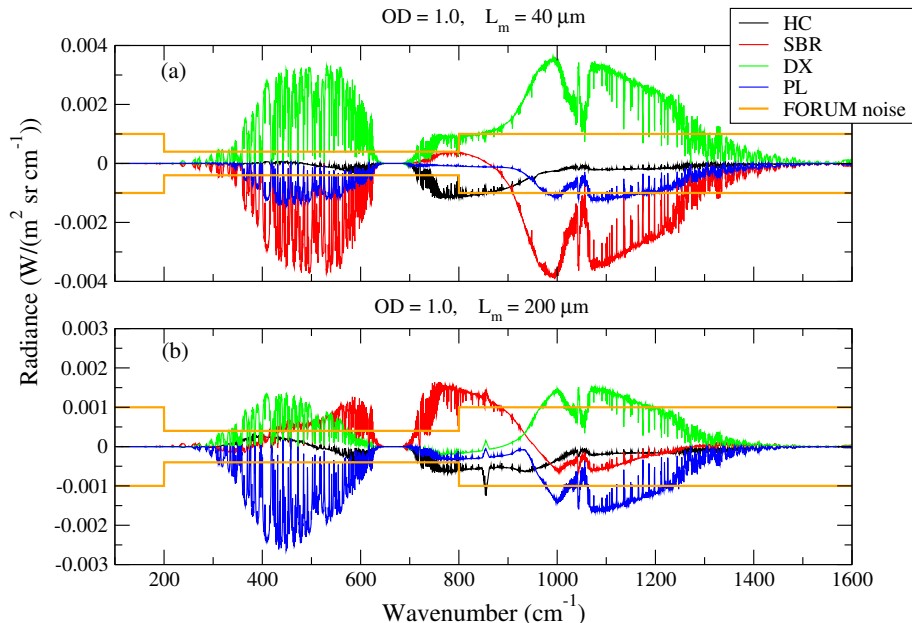

**Figure 2.** Differences between radiances obtained with perturbed and reference (uniform) habit mixture. The curve colors refer to the specific habit whose fraction has been increased by 0.6. Panel (a) refers to the cloud with $L_m = 40$ $\mu$m, while panel (b) to the cloud with $L_m = 200$ $\mu$m.

position of the cloud. For the radiative transfer calculations we directly set the total cloud optical depth to the value for which we want to assess the retrieval procedure. The ERA5 surface type information allows us to associate to each specific scenario a surface spectral emissivity profile extracted from Huang et al. (2016).

More specifically, we select three different scenarios from ERA5: a scenario at mid-latitudes [48.5 N, 81.5 E] (2019/08/05,
12:00:00 UTC), a tropical scenario [6.5 S, 64.5 E] (2019/08/05, 00:00:00 UTC) and a polar scenario [75.5 S, 123.5 W] (2019/08/14, 12:00:00 UTC). The ice cloud layers are placed in the following height ranges: 6–9 km at mid-latitudes, 14–17 km at the tropics, 4–6 km in the polar scenario. Clouds are considered both vertically and horizontally homogeneous. The surface height is assumed to coincide with the average mean sea level (AMSL) but for the polar scenario that is located on the Antarctic Plateau, thus the surface is assumed to be 3000 m above the AMSL. In the three mentioned scenarios, surface
temperatures are of 10, 40 and -60°C, respectively.

For each of the three selected scenarios, we explore various cloud configurations by artificially changing the cloud OD, $L_m$ and habit mixture fractions ($p_1$, $p_2$, $p_3$, $p_4$). We set alternatively OD equal to 0.1, 0.5, 1, 2 and 4. For each of these OD values, $L_m$ takes the values of 40, 100, 200, 300 and 400 $\mu$m. Then, for each couple (OD, $L_m$), we assume, in turn, the five scenario-dependent habit mixtures ($p_1$, $p_2$, $p_3$, $p_4$) listed in Table 1. In summary, we explore 125 different ice cloud configurations for
each atmospheric scenario, for a total of $125 \times 3 = 375$ test cases. For each of these cases, we simulate the high-resolution upwelling spectral radiance at the top of the atmosphere and convolve it by the FORUM instrument spectral response function

already mentioned in Sect. 2.4. Spectrally un-correlated pseudo-random noise compliant with FORUM requirements (see Sect. 2.4) is then added to the radiance. Noisy and noise-free radiances are both stored for later use in simulated retrievals.

**Table 1.** Summary of the ice crystal habit mixtures investigated in this work.

| | $p_1$ (HC) | $p_2$ (SBR) | $p_3$ (DX) | $p_4$ (PL) | $p_1$ (HC) | $p_2$ (SBR) | $p_3$ (DX) | $p_4$ (PL) | $p_1$ (HC) | $p_2$ (SBR) | $p_3$ (DX) | $p_4$ (PL) |
|---|---|---|---|---|---|---|---|---|---|---|---|---|
| | Mid-latitude | | | | Tropics | | | | Polar | | | |
| a) | 0.80 | 0.10 | 0.05 | 0.05 | 0.80 | 0.10 | 0.05 | 0.05 | 0.80 | 0.10 | 0.05 | 0.05 |
| b) | 0.10 | 0.80 | 0.05 | 0.05 | 0.10 | 0.80 | 0.05 | 0.05 | 0.05 | 0.80 | 0.05 | 0.10 |
| c) | 0.05 | 0.10 | 0.80 | 0.05 | 0.05 | 0.05 | 0.80 | 0.10 | 0.05 | 0.05 | 0.80 | 0.10 |
| d) | 0.05 | 0.05 | 0.10 | 0.80 | 0.10 | 0.05 | 0.05 | 0.80 | 0.05 | 0.05 | 0.10 | 0.80 |
| e) | 0.20 | 0.30 | 0.30 | 0.20 | 0.30 | 0.25 | 0.25 | 0.20 | 0.40 | 0.10 | 0.40 | 0.10 |

## 3.2  Self consistency of the retrieval algorithm and convergence error

First, we study the self-consistency of the inversion procedure and assess the convergence error by applying the retrieval to the set of noise-free synthetic measurements described above. In these test retrievals, the a priori estimate of the state vector $\mathbf{x}_a$ is set equal to the reference (or *true*) state that has been used to generate the synthetic measurement, thus, no bias is expected on the solution of the retrieval and the expectation value of the cost function in Eq.(18) is zero. To minimise the effect of the constraint provided by the a priori, we use very large a priori errors: 100% relative error for OD and $L_m$, and error equal to 1 in the habit fractions $p_1, \ldots, p_4$. The initial guess of the retrieval is set up as follows: OD and $L_m$ are obtained by applying a random perturbations with relative amplitudes of 100 and 50% to the respective true values, while the four habit fractions are set equal to 0.25 (uniform habit distribution).

Figure 3 shows the differences ($\delta$, black dots) between the true and the retrieved habit fractions for all the considered scenarios. Panels (a) to (d) refer to the four crystal analysed habits HC, SBR, DX and PL, respectively. The plots also show the $1\sigma$ retrieval error estimated as the mapping of the measurement noise onto the solution of the retrieval (blue lines). The largest retrieval errors occur for the polar scenarios where clouds are very close to the surface ($\approx$1 km above the ground) that has a temperature of -60 °C, very similar to that of the cloud itself. In these conditions, the measurement hardly discriminates between the surface and the cloud contributions to the upwelling radiance.

At the first run of these test retrievals, we found that about 12% of the cases were missing the convergence to the absolute minimum of the cost function. This issue was attributed to the ill-posed nature of our inversion problem. In general, the relatively weak sensitivity of the measurements to the retrieval parameters decreases the curvature (i.e. makes "flat") the hyper-surface of the CF of which we want to find the absolute minimum. Moreover, the measurement noise and the forward model errors introduce a sort of "roughness" in the CF hyper-surface. Finally, if some parameters included in the state vector are correlated to each other, narrow "canyons" may also characterize the hyper-surface of the CF (Transtrum et al., 2011; Ridolfi

and Sgheri, 2013). All-together, these elements make the inversion problem ill-posed, and the search for the absolute minimum of the CF becomes a challenging task. The Gauss-Newton method modified with the LM damping, being based on the CF gradient may get trapped on a secondary, local minimum of the CF, as it actually happened in our first retrieval attempts. A solution could have been to use a stochastic method in place of the LM to find the CF minimum. For example, in Ridolfi and Sgheri (2009), the simulated annealing method was used to find the absolute minimum of a CF specifically designed to find the optimal strength of the height-dependent Tikhonov regularization applied to the retrieval of vertical profiles from limb sounding spectral measurements. The simulated annealing, however, as other stochastic minimization methods, requires thousands of evaluations of the CF. In our case, the evaluation of the CF implies the evaluation of the forward model, thus a computationally heavy operation. This feature clearly makes stochastic methods inadequate for our application.

We managed to overcome almost completely this convergence issue by repeating a few times the retrieval starting from slightly different initial guesses of the state vector and by selecting, a posteriori, the solution corresponding to the smallest value of the cost function. With this strategy, only 4 out of the 375 retrievals actually miss the absolute minimum of the CF.

It is worth to note that in the simulated retrievals, the few cases with a large convergence error are easily detected because in these cases a difference much larger than the error bar exists between the retrieved and the true parameter values. In real data analysis the problem may be harder to identify. A strategy could be to compare the retrieved parameter values and the achieved CF minimum with the respective accumulated statistics, and to treat the outliers as "suspicious" cases. In these cases, restarting the retrieval from different a priori parameter estimates could be beneficial, as it proved to be in the test retrievals presented here. Also a visual inspection of the residuals of the fit for some selected suspicious cases could help to diagnose the problem and to find a workaround.

Figure 3 refers to the results finally obtained from our test retrievals. We see that almost the totality of the retrieved habit fractions differ from their true values by amounts much smaller than the retrieval error predicted on the basis of the measurement noise, thus the convergence error is generally negligible.

To ease the comparison of the new proposed scheme with the more classical approaches that assume fixed habit mixtures, using the retrieved values of $L_{\mathrm{m}}$, the habit fractions $p_1, \ldots, p_4$ and their retrieval error estimates $\Delta L_{\mathrm{m}}$ and $\Delta p_1, \ldots, \Delta p_4$, we also computed $D_e$ with equation (7) and an estimate of its error $\Delta D_e$ as:

$$\Delta D_e = \sqrt{\left| \frac{\partial D_e(\mathbf{p}, L_{\mathrm{m}})}{\partial L_{\mathrm{m}}} \right|^2 \Delta L_{\mathrm{m}}^2 + \sum_{h=1}^{4} \left| \frac{\partial D_e(\mathbf{p}, L_{\mathrm{m}})}{\partial p_h} \right|^2 \Delta p_h^2} \tag{22}$$

Panel (a) of Fig. 4 shows (red symbols) the differences between true and retrieved effective particle diameters for each of the considered scenarios. As usual, the blue line represents the retrieval error estimated on the basis of the propagation on the measurement noise onto the solution of the retrieval. Panel (b) is analogous to panel (a) but refers to the retrieved OD. Finally, panel (c) of Fig. 4 shows the values of the normalized cost function $\chi_{\mathrm{N}}^2$ at the beginning (black) and at the end (orange) of the retrieval iterations. We see that the differences between retrieved and true quantities, in general, are much smaller than the retrieval error due to measurement noise, moreover, the final $\chi_{\mathrm{N}}^2$ is much smaller than 1. This behavior proves the self

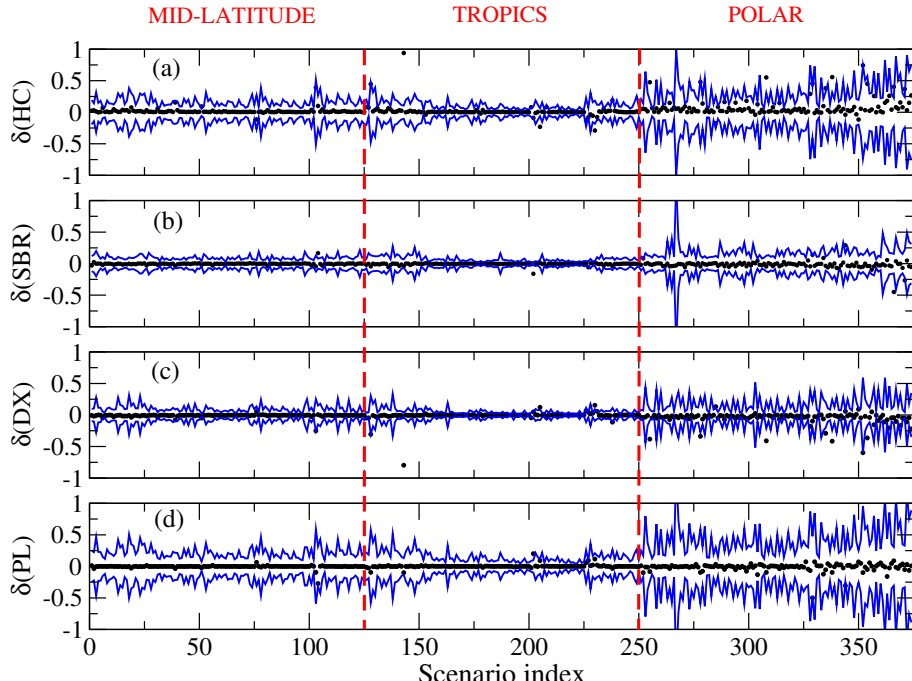

**Figure 3.** Test retrievals based on noise-free measurements. The black symbols represent the differences ($\delta$) between true and retrieved habit fractions. For reference, the blue lines represent the retrieval errors due to measurement noise. The panels from (a) to (d) refer to the four crystal habits considered, as indicated in the vertical axis labels.

consistency of the inversion algorithm, and shows that the convergence error is actually much smaller than the retrieval error expected on the basis of the measurement noise.

### 3.3 Retrievals from FORUM-like measurements

We repeated the test retrievals presented in Sect. 3.2 starting from the synthetic radiances affected by measurement noise as anticipated for the FORUM sensor. In this case, to detect if the retrieval converged to a secondary minimum of the cost function, the retrievals achieving values of $\chi_N^2 > 1.1$ were repeated several times, starting from slightly different initial guess parameters. More specifically, the retrieval was repeated 10 times, each time starting from $L_m$ and OD values extracted from a $5 \times 2$ matrix (5 $L_m$ values $\times$ 2 OD values). Finally, we selected the results of the run that ended with the smallest $\chi_N^2$. This approach was

selected as a reasonable compromise between the need to avoid convergence to secondary minima of the cost function, and the computation time required.

Figures 5 and 6 show the results of this set of test retrievals, using the same lines and colours as in Figures 3 and 4, respectively. Note that most of the parameter differences fall within the measurement noise error boundaries but in some specific cases, mostly concentrated in polar atmospheric scenarios, confirming, as explained in Sect. 3.2, that the retrieval of

cloud parameters from polar scenarios is a particularly challenging task.

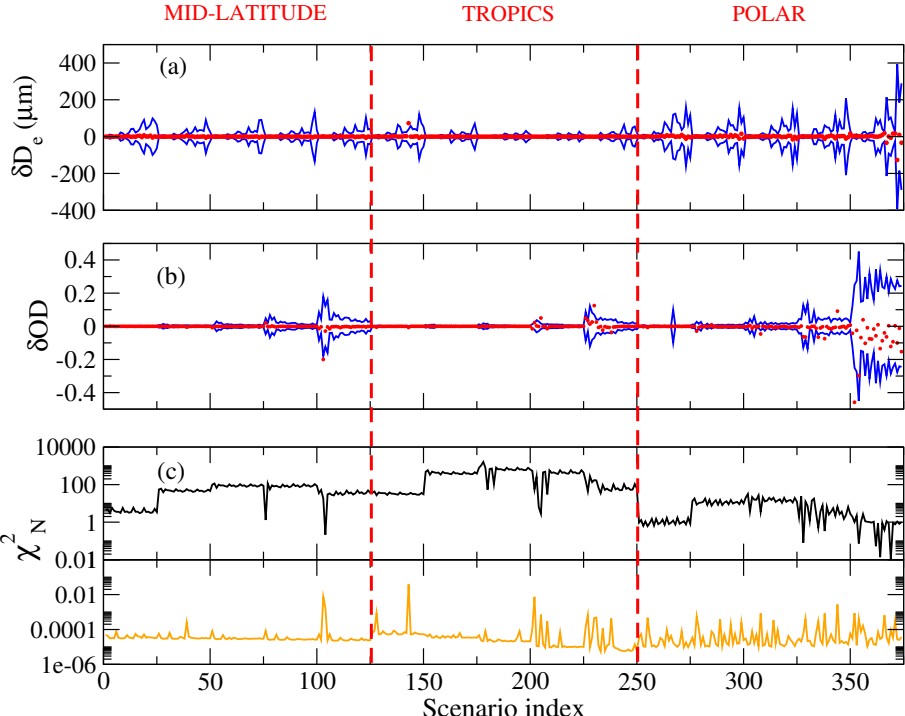

**Figure 4.** Retrieval from noise-free measurements. Panel (a) shows the differences between true and retrieved effective particle diameters (red symbols) and the retrieval error estimated on the basis of the propagation of the measurement noise onto the solution (blue). Panel (b) is analogous to panel (a) but refers to OD. Panel (c) shows the values of the normalized cost function $\chi^2_N$ at the beginning (black) and at the end (orange) of the retrieval.

The results reveal that the minimisation algorithm is able to identify a good minimum of the cost function, with values of $\chi^2_N < 1.1$ for 371 out of the 375 scenarios considered in this analysis. In most of the cases, the minimum achieved corresponds to $L_m$ and OD values differing from their true values by amounts comparable or smaller than the error due to measurement noise. Unfortunately, the same does not always hold for the retrieved crystal habit fractions: in some specific scenarios their retrieved values differ from the true values over the noise error boundaries and, still, the achieved $\chi^2_N$ is reasonably small. This means that a $\chi^2_N$ minimum comparable with the absolute minimum has been found, corresponding to a combination of habit fractions different than the real one. Again, this effect indicates that for some specific scenarios the inversion is strongly ill-conditioned, i.e. several solutions exist, providing almost the same minimum of the CF.

## 4 Errors due to uncertainties in the assumed model parameters

In principle our SACR code has the capability to retrieve simultaneously atmospheric, surface and cloud parameters. The retrieval of cloud parameters, however, on its own is already a challenging task. In particular, in Sect.s 3.2 and 3.3, we have

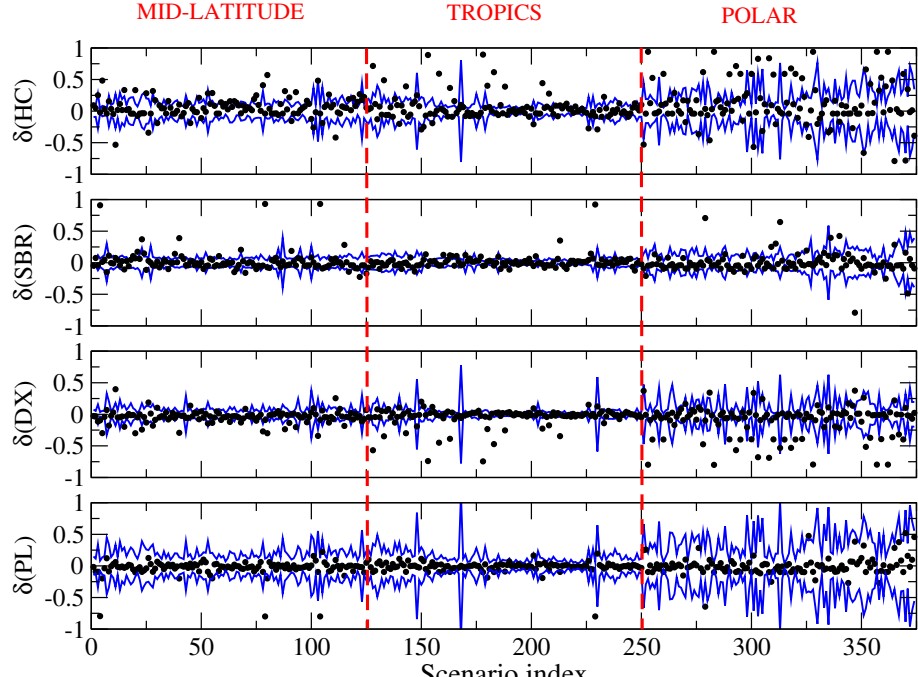

**Figure 5.** Test retrievals based on FORUM synthetic radiances affected by measurement noise. The black symbols represent the differences ($\delta$) between true and retrieved habit fractions. For reference, the blue lines represent the retrieval errors due to measurement noise. The panels from (a) to (d) refer to the four crystal habits considered, as indicated in the vertical axis labels.

seen that the retrieval of the habit fractions, the cloud $L_m$, and the OD already turns out to be an ill-conditioned inversion in specific atmospheric / cloud scenarios. For this reason, we do not retrieve simultaneously atmospheric, surface and cloud parameters. Indeed, the test retrievals presented above assume as *known* the atmospheric temperature and gas profiles, surface temperature and spectral emissivity, and the cloud top and bottom heights (CTH and CBH). An error on these parameters will cause a *forward model error* that, in turn, will propagate onto the retrieved cloud parameters as an additional uncertainty to be combined with the error due to the measurement noise.

In this Section, we examine the cloud parameter errors due to the uncertainties on the assumed temperature ($\mathbf{T}$) and water vapor ($\mathbf{WV}$) profiles, on the surface temperature ($T_s$) and spectral emissivity ($\epsilon$), on CTH and CBH. These are expected to be the most relevant error components affecting the retrieved cloud parameters. Let $\mathbf{b}$ be the vector including all the relevant assumed model parameters affected by uncertainty. In our case:

$$\mathbf{b} = (\mathbf{WV}, \mathbf{T}, \epsilon_{\mathbf{s}}, T_s, \text{CBH}, \text{CTH}). \tag{23}$$

Let $\mathbf{S}_b$ be the error covariance matrix of $\mathbf{b}$. The error $\mathbf{S}_b$ maps (Rodgers, 2000) into a forward model error covariance $\mathbf{S}_f$ as:

$$\mathbf{S}_f = \mathbf{K}_b \mathbf{S}_b \mathbf{K}_b^T, \tag{24}$$

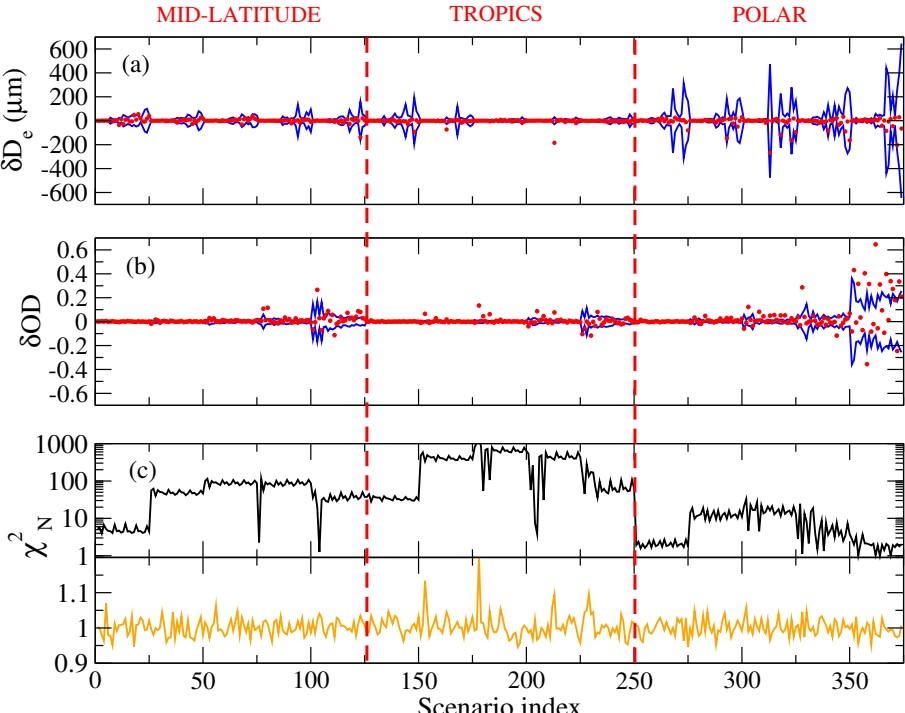

**Figure 6.** Test retrievals based on FORUM synthetic radiances affected by measurement noise. Panel (a) shows the differences between true and retrieved effective particle diameters (red symbols) and the retrieval error estimated on the basis of the propagation of the measurement noise onto the solution (blue). Panel (b) is analogous to panel (a) but refers to OD. Panel (c) shows the values of the normalised cost function $\chi_N^2$ at the beginning (black) and at the end (orange) of the retrieval.

where $\mathbf{K}_b$ is the Jacobian matrix containing the derivatives of the forward model with respect to the parameters of the vector $\mathbf{b}$. The forward model error covariance $\mathbf{S}_f$ adds up to the measurement error covariance matrix $\mathbf{S}_y$ to build the total error covariance matrix $\mathbf{S}_y'$ of the difference $\mathbf{y} - \mathbf{f}(\mathbf{x})$ appearing in the cost function Eq. (18):

$$\mathbf{S}_y' = \mathbf{S}_y + \mathbf{S}_f = \mathbf{S}_y + \mathbf{K}_b \mathbf{S}_b \mathbf{K}_b^T \tag{25}$$

To evaluate $\mathbf{S}_b$ we proceed as follows. For temperature and water vapour profiles, we assume the error profiles given in Table 2.
These are the background errors assumed at the UK MetOffice for the assimilation of the IASI (Infrared Atmospheric Sounding Interferometer) measurements in their operational numerical weather prediction system. Consistently, we assume an error of 1 K on surface temperature and an error of 0.1 on surface spectral emissivity. For CTH, we assume an error of 0.8 km. This is more or less the uncertainty that one may expect if CTH was derived from FORUM spectral radiance, using e.g. the $CO_2$-slicing method proposed in Holz et al. (2006) and Taylor et al. (2019). For CBH, we assume an error of 0.5 km as resulting
from the analysis of Di Natale et al. (2020). The off-diagonal elements of $\mathbf{S}_b$ that correlate the grid points of the same vertical

profile are set assuming a correlation $c_{ij}$ between the elements at heights $z_i$ and $z_j$, given by:

$$c_{ij} = \exp \left[ - \frac{|z_i - z_j|}{\eta} \right],$$

(26)

where $\eta$ is the correlation length that we take equal to 5 km.

**Table 2.** Assumed errors for temperature and water vapour profiles.

| z(km) | $\Delta$T(K) | $\Delta$(WV)(%) |
|-------|-------|-------|
| 120.0 | 1.4 | 10.0 |
| 32.1 | 1.4 | 10.0 |
| 32.0 | 0.6 | 10.0 |
| 15.1 | 0.6 | 10.0 |
| 15.0 | 0.6 | 30.0 |
| 8.1 | 0.6 | 30.0 |
| 8.0 | 0.5 | 30.0 |
| 2.6 | 0.5 | 30.0 |
| 2.5 | 0.5 | 3.0 |
| 0.0 | 0.5 | 3.0 |

For each of the scenarios presented in Sect. 3.3, we repeated the retrieval using $\mathbf{S}'_y$ instead of $\mathbf{S}_y$. Figure 7 summarizes the
errors in the retrieved habit fractions with the same notation used in Figs. 3 and 5. In Fig. 7, the dashed light-blue lines represent
the retrieval errors estimated on the basis of the total error covariance matrix of the retrieval. This error is to be compared to
the noise component of the retrieval error (dashed blue line) that was estimated in the retrievals presented in Sect. 3.3. Clearly,
the total error is larger than the single error component caused by the measurement noise, however its value is still reasonable,
allowing one to build a climatology of ice crystal habits.

**5   Impact of crystal habit assumption on flux calculations**

In the retrieval of cloud properties, the habit distribution is usually assumed unless simultaneous local measurements are
available. For example, for the validation of the Infrared Imaging Radiometer (IIR) onboard the Cloud-Aerosol Lidar and
Infrared Pathfinder Satellite Observation (CALIPSO), Sourdeval et al. (2013) used the crystal habits measured by a lidar and
an infrared radiometer installed onboard of the Falcon-20 aircraft during two measurement campaigns in 2007 and 2008.
On the contrary, in King et al. (2004) the authors retrieve ice and liquid cloud optical and micro-physical properties from the
measurements of the Moderate Resolution Imaging Spectroradiometer (MODIS) Airborne Simulator (MAS), a simulator of the
MODIS satellite flown on board of the National Aeronautics and Space Administration (NASA) ER-2 high-altitude research
aircraft over Alaska as part of the First International Satellite Cloud Climatology Project (ISCCP) Regional Experiment–Arctic

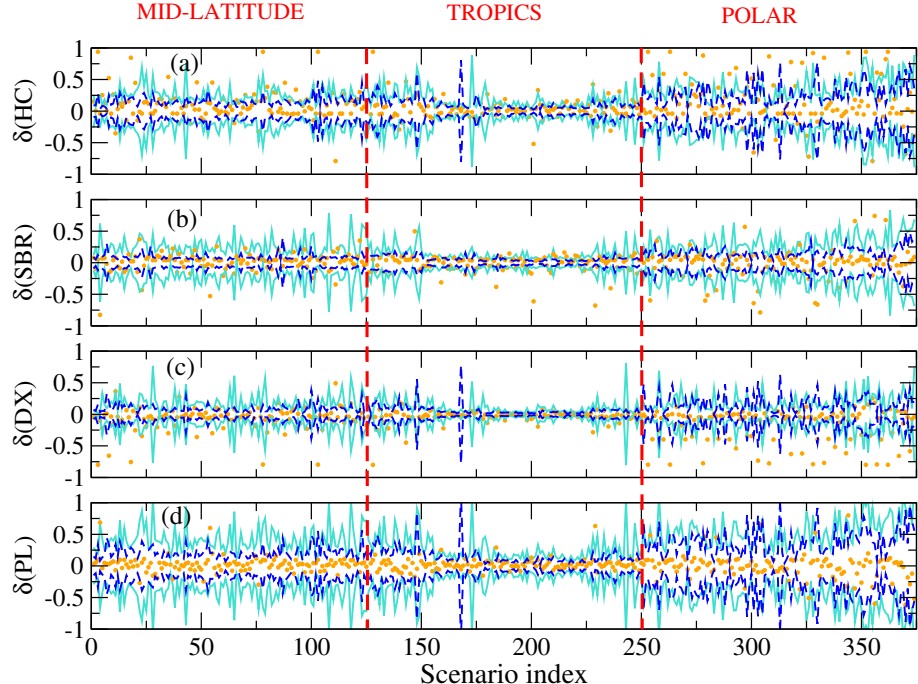

**Figure 7.** Test retrievals based on FORUM synthetic radiances, with total errors $\mathbf{S}_y'$ taken into account. The orange symbols represent the differences between true and retrieved habit fractions. The dashed blue lines represent the retrieval errors due to measurement noise. The dashed light-blue lines represent the total retrieval error. Panels (a) to (d) refer to the four crystal habits considered, as indicated in the vertical axis labels.

Clouds Experiment (FIRE ACE). In this case the authors assumed the particle habit mixture resulting from a statistical analysis of local in-situ measurements.

Baum et al. (2005b) and Baum et al. (2007) developed specific databases of both band-integrated and spectrally-resolved cloud radiative properties to be used in the retrievals from the MODIS multispectral imager and hyperspectral radiometers (like the Atmospheric Infrared Sounder, AIRS). In these works, the size and habit distributions are estimated from several previous field campaigns as described in Baum et al. (2005a).

To improve the agreement between cloud properties retrieved from MODIS and from the infrared spectral band retrievals, the collection number 5 (C5) of crystal habit properties was improved, leading to the collection 6 (C6). C6 uses an improved ice scattering model developed on the basis of the results of Holz et al. (2016). These authors, using a month of collocated A-Train observations, highlighted a systematic bias between ODs derived from MODIS C5 and CALIOP Version 3 (V3) unconstrained retrievals, for OD < 3. The single-scattering ice properties of C5 were found to be responsible for this bias. To overcome this inconsistency, the collection C6 was built assuming a modified gamma distribution to compute the average scattering properties of a single crystal habit of severely roughened aggregated columns.

Iwabuchi et al. (2016) developed a cloud retrieval algorithm based on the optimal estimation and tested their approach using ten thermal infrared bands measured by MODIS. In this case, column aggregate crystal habits with very rough surfaces were assumed for consistency with the MODIS C6 cloud product.

A crystal habit distribution is also assumed in the cloud property retrievals from AIRS data presented in Li et al. (2005).

    Using a variational algorithm, Delanoë and Hogan (2010) combine data from spaceborne radars, lidars and infrared radiometers on the "A-Train" satellites, to retrieve ice cloud properties. In this case, the authors assume the crystal habit distribution and evaluate the implied error by exploiting the a priori radiance error presented in Cooper et al. (2003).

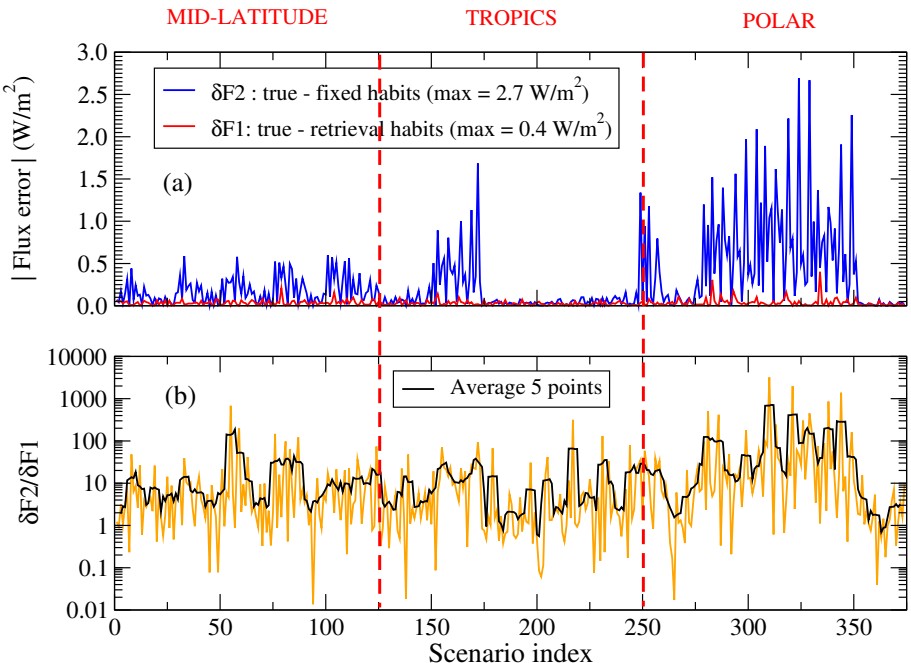

**Figure 8.** Panel (a): differences between the OLR fluxes calculated by fitting the habit weights ($\delta F1$, red curve) and by fixing the habit crystals assuming the distribution by King et al. (2004) for MODIS/MAS retrievals ($\delta F2$, blue curve). Panel (b): ratio between $\delta F2$ and $\delta F1$.

    For the MODIS/MAS retrievals, King et al. (2004) assume a habit distribution made of a mixture of plates, rosettes, and

hollow columns when the maximum diameter is < 70 $\mu$m, and a mixture of plates, rosettes, hollow columns and aggregates when the maximum diameter is > 70 $\mu$m (Baum et al., 2000).

    To estimate the errors that the assumption of a given habit distribution would generate in our simulation experiment, starting form the synthetic measurements presented in Sect. 3.3, we carried out the retrieval using two different configurations. In the first case (1), we assume the habit distribution equal to the distribution of King et al. (2004) and we retrieve only the $L_{\mathrm{m}}$ and OD

cloud parameters. In the second case (2), we still start the retrieval using the distribution of King et al. (2004), and we retrieve $L_{\mathrm{m}}$, OD and the habit fractions $p_1, \ldots, p_4$. In both cases, the retrieval assumes the correct surface properties, atmospheric state and the cloud top position.

We then computed the OLR flux by integrating the spectral radiance over the solid angle and over the 100-1600 cm$^{-1}$interval. Figure 8 shows the absolute differences between the OLR fluxes computed from the spectra simulated at the last iteration of the retrievals (1) and (2), and the "real" spectra, i.e. the spectra simulated with the atmospheric, cloud and surface states assumed to be the local *truth*. The red line refers to the retrievals (2), where also the habit mixture is retrieved, while the blue line refers to the retrievals (1) assuming the fixed King et al. (2004) habit mixture. We clearly see that the flux error caused by the erroneous assumptions of the crystal habits mixture largely exceeds the error that we have when also the habit fractions are retrieved. With the specific observational scenarios considered here, the error caused by the assumed habits mixture may amount up to 2.7 W/m$^2$. This means that if the assumed habit mixture was uniformly biased all over the globe, it could cause a energy flux error comparable to the global Earth radiation imbalance currently estimated to be in the range from 0.6 to 1.2 W/m$^2$ (Wild, 2020).

Note that in the inversions of type (1) assuming a given habit mixture, the minimization procedure may use cloud parameters $L_m$ and OD to compensate for the erroneous assumption of ice habit mixture. As shown in our example, however, this compensation is only partial as the flux error turns out to be quite relevant. Of course, in this case the compensation effect produces a bias in the retrieved values of $L_m$ and OD. When also the habit fractions are retrieved, the flux error turns out to be significantly reduced, and the bias of $L_m$ and OD disappears.

## 6 Conclusions

Ice crystal habit distributions are usually assumed in the retrieval of cloud properties from atmospheric radiances. This assumption, however, introduces biases in the statistics of the estimated cloud parameters and in the derived outgoing longwave radiation fluxes. Since the climatologies of cloud parameters and of outgoing longwave radiation fluxes are also used to tune and constrain the radiative parametrisations included in global climate models, the above mentioned biases also affect the accuracy of climate predictions.

To avoid these biases, we introduce in our SACR inversion scheme the capability to retrieve the ice particle habit fractions. Within the inversion scheme, the forward model simulates the radiative transfer through the cloudy atmosphere using rigorous formulas for the bulk cloud optical properties of a mixture of ice crystals with various habits. These formulas can be evaluated with a reasonable computing time, exploiting the new habit-specific lookup tables that we built for this purpose. These lookup tables contain the integrals appearing in the formulas of the optical properties, tabulated as a function of the parameter $L_m$ that characterises the particle size distribution.

We verify the self consistency of the developed inversion scheme and assess the performance of the habit fraction retrievals on the basis of simulated spectral radiance observations of the forthcoming ESA FORUM space mission. The simulations utilise 375 different observational scenarios including various ice cloud configurations, from Tropical to Polar latitudes. In our test setup, the state vector of the retrieval includes the cloud optical depth OD, the parameter $L_m$ and four habit fractions $p_1, \ldots, p_4$. We evaluate both the retrieval error due to the expected measurement noise, and the error induced by assumptions of fixed forward model parameters: temperature and water vapour profiles, surface temperature and emissivity, cloud top and

bottom heights. On average, the total retrieval error on the habit fractions is around 0.2, with the actual retrieval error depending significantly on the specific atmospheric scenario considered.

The results of our test retrievals highlight that the inversion of habit fractions is ill-conditioned. In fact, especially at polar latitudes, we find cases in which the retrieval converges to very good (low) minimum of the cost function, with habit fraction values differing (far beyond error bars) from their reference value used in the generation of the simulated measurements. In most of the cases this issue is solved by starting the retrieval using a different initial guess of the state parameters. Despite of that, we recognise that detection and solution of this issue may be challenging when real measurements will be processed.

For the selected measurement scenarios, we also evaluate the error on the total OLR energy flux caused by the choice of using pre-defined habit mixtures instead of retrieving the habit fractions. For some polar scenarios, this error may be as large as 2.7 $W/m^2$. This means that if the assumed habit mixture was uniformly biased all over the globe, it could cause a energy flux error comparable to the global Earth radiation imbalance currently estimated to be in the range from 0.6 to 1.2 $W/m^2$ (Wild, 2020).

In perspective, we plan to apply the developed retrieval scheme to the large dataset of ground based spectral radiance measurements collected in Antarctica by the Radiation Explored in the Far-InfraRed - Prototype for Applications and Development (REFIR-PAD), a Fourier spectroradiometer deployed by our institute since December 2011 (Palchetti et al., 2015). These measurements are complemented by co-located backscattering/depolarization lidar measurements permitting to estimate the cloud boundaries, and by the statistics of ice crystal shapes determined from an ice camera deployed on the same site. Potentially, the synergy of these measurements could allow to validate the inversion method presented in this work on the basis of real data and, at the same time, to build a climatology of Antarctic cloud ice crystal habits, corroborated by local in situ measurements.

The presented method is designed for application to the nadir spectral measurements of the forthcoming FORUM and PREFIRE satellite missions, with the potentiality to derive accurate statistics of cloud properties that are extremely relevant for climate studies. In principle, the method could also be applied to the current satellite measurements (such as those of IASI) that are limited to the mid-infrared region. However, in this latter case, it is uncertain whether these measurements contain sufficient information to disentangle the contributions of the various ice crystal shapes to the upwelling spectrum.

*Author contributions.* Conceptualization, GDN; methodology, GDN, MR; software, GDN; validation, GDN; formal analysis, GDN; investigation, GDN, MR; resources, MR, GDN; data curation, GDN; writing–original draft preparation, GDN; writing–review and editing, all; visualization, all; supervision, MR, LP; funding acquisition, GDN, LP. All authors have read and agreed to the published version of the manuscript.

*Competing interests.* The authors declare that they have no conflict of interest.

*Acknowledgements.* The authors gratefully acknowledge the Italian PNRA (Programma Nazionale di Ricerche in Antartide) and, specifically, the FIRCLOUDS (Far-Infrared Radiative Closure Experiment For Antarctic Clouds) project, which provided funding support for the first author (contract no. 0048159 6.7.2018). This research has also been supported by the European Space Agency (ESA–ESTEC contract no. 4000137321/22/NL/AD) and by the Italian Space Agency (ACCORDO ATTUATIVO ASI-CNR-INO FORUM SCIENZA n. 2019-20-HH.0).

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
