# Peer review of "A new approach to the crystal habit retrieval from far infrared spectral radiance measurements"

_EGUsphere, 2023_

## Author Comment (AC1)

We thank the reviewer for his constructive comments. We believe the comments helped us to improve the overall quality of the manuscript. Please find below our answers to the comments, with indication of the changes made in the revised manuscript. For clarity, we also include the original reviewer's comments in **bold**.

Manuscript : **"A new approach to the crystal habit retrieval from far infrared spectral radiance measurements"** , Di Natale et al. 2023

**This paper presents an inversion algorithm for retrieving the shapes of ice cloud particles using far infrared spectral radiance measurements. The algorithm's performance is evaluated using simulated measurements from the new far infrared sensor, Far-infrared Outgoing Radiation Understanding and Monitoring (FORUM), in various scenarios: tropics, mid-latitudes, and polar regions. This study is a continuation of a previous paper by Di Natale and Palchetti (2022) that focuses on the algorithm's development and validation.**
**Overall, the results indicate a successful convergence of the inversion algorithm in tropical and mid-latitude scenarios. However, some challenges were encountered in the polar scenario, particularly when considering retrieval affected by FORUM measurement noise. Notably, there were significant differences in the simulated outgoing longwave radiation when using pre-defined fixed shapes compared to retrieved shapes. These findings have implications for improving ice cloud parameterization and enhancing our understanding of ice particle habits in different locations.**

**While the paper is logically structured and well-organized, I have a few suggestions to enhance its clarity:**

**1.Figure 1: Why does the difference between the radiance computed with the rigorous approach and the approximation increase with OD before OD < 2.0 but decrease when OD > 2.0? The author should provide an explanation for this trend.**

In the revised version of the paper, we replaced Fig. 1 with a simpler and more informative version. The revised Figure 1 is made of two panels (a) and (b), referring to the computations made with $L_m$=200µm and $L_m$=40µm, respectively. Each panel shows the radiance differences obtained for several values of OD, spanning a broad range, from 0.001 to 150.
Explaining the behavior of these differences on the basis of the radiative transfer equation and of the expressions used to compute the various contributing elements (see Di Natale et al. 2020) is a very complicated issue. Despite of this complication, we see that the asymptotic behavior of the differences, for OD $\to$ 0 and for OD $\to \infty$ is reasonable. Specifically, for OD $\to$ 0, i.e. for ice amounts getting closer and closer to zero, the cloud effect on the upwelling spectral radiance must get closer and closer to zero, thus the two compared methods should provide the same result, as confirmed by the lines of Fig. 1 corresponding to the smallest OD values. Conversely, in the presence of a very opaque cloud (OD>>1), the radiance should depend uniquely on the absorption and scattering processes occurring at the cloud top. Therefore, we expect the differences between the radiance predicted by the two methods to approach a wavenumber - dependent asymptotic value that does not change for any further increase of cloud OD. This behavior is actually confirmed looking at the lines of Fig. 1 that correspond to OD $\geq$ 30, they almost overlap. Note that the differences between the two methods increase for increasing OD, reach a maximum amplitude for OD ~ 2, then decrease to their asymptotic value achieved for OD >> 1.
In Sect. 2.4 of the revised paper, we included this explanation of the asymptotic behaviors of the observed differences between the two methods compared.

**2. Section 2: I am wondering which database of the optical properties of ice crystals was used in this study. Yang et al. (2013) was continuously updated based on the improvement of computational techniques. For example, Bi and Yang (2017) updated the database based on the invariant imbedding T-matrix method (Bi and Yang, 2014) and improved ray tracing technique for absorbing particles.**

We are aware about the latest release of the single scattering properties databases by Bi and Yang (2017), however, we preferred to use the databases of Ping Yang et al. (2013) to allow for the intercomparability of this new rigorous approach presented, with the approximated method of Di Natale and Palchetti, (2022). For the present work this choice does not represent a limitation as we are dealing only with simulated measurements. We fully agree, however, that for the analysis of real measurements we will have to use the most recent release of the single scattering properties databases. However, we added the reference mentioned by the reviewer in the text at line 116.

In the revised manuscript, we added a statement to justify our choice. Again, the statement was added in Sect. 2.4.

**3. Section 5, Lines 399-410: The author conducted test retrievals using the predefined habits of King et al. 2004 and the retrieved habits in this study, assuming that the atmospheric, surface, and cloud parameters were known. However, if the cloud parameters were also influenced by the habits in the retrieval when considering simultaneous retrieval of cloud parameters, I wonder if the value of 2.7 W/ would be lower. Additionally, it is possible that the difference caused by the habits would be partially compensated by adjusting the cloud parameters.**

Probably this part was not clear in the original manuscript. In the revised manuscript, we rephrased the related paragraph in Sect. 5, to better explain results of the test we carried out.

Starting from the synthetic measurements generated for the analysis presented in Sect. 3.3, here we carry out the retrieval of cloud parameters (OD and $L_m$) with two different approaches: in the first case (a) we assume the particle habit distribution to be known and equal to the distribution of King et al. 2004. In the second case (b), along with OD and $L_m$, we also retrieve the habit fractions $p_1$, …, $p_4$ introduced in Sect. 2.1. In both cases (a) and (b), the atmospheric and surface states as well as the cloud top are assumed as exactly known.
In figure 8, we finally compare the OLR fluxes computed from the atmospheric and cloud states obtained at the end of the retrievals (a) and (b), to the fluxes obtained from the "true" atmospheric and cloud states assumed for the generation of the synthetic measurements.

In conclusion, we agree that the retrieval may use cloud parameters, such as OD and $L_m$, to compensate for an eventually erroneous assumption of the ice particle distribution, however this compensation is actually free to happen also in our retrievals of case (a). Thus, the possible reduction of the flux error by compensation with the retrieved cloud parameters is already included in the results presented in figure 8. From the figure it is clear that, the compensation effect reduces only partially the flux error as this latter is still larger for case (a) than for case (b). The two panels in the figure below show the differences between cloud parameters OD and $L_m$ retrieved in the cases (a) and (b), highlighting the amplitude of the systematic compensation effect mentioned above. For

brevity, we decided for not including these figures in the revised paper, however in the revised paper we expanded the discussion of the results.

Note that, in principle, when retrieved simultaneously, also the parameters relating to temperature or water vapor vertical profiles or to surface temperature or surface spectral emissivity could compensate for an erroneous cloud ice size distribution assumption. The effectiveness of this type of compensations in reducing the OLR flux error, however, is expected to be much lower as compared to that of cloud parameters. This is because the spectral fingerprints of atmospheric and surface parameters have a shape substantially different from that due to cloud parameters.

[Figure]

**4.In Figure 1, the y-axis tick labels of "0,001" should be "0.001". This typo also exists in Figures 4, 6, 7, and 8.**

Corrected.

**5.Line 278: "the a priori..." should be corrected.**

Corrected.

**6.Line 324: "...a part 4 out..." should be corrected.**

Corrected.

**7.Line 319: "Fig.s 3 and 4..." should be corrected.**

Corrected.

**8.Line 410: "(0.6+-0.4)W/m^2 Wild et al. (2013)" should be corrected.**

Corrected.

References:

Di Natale, G., & Palchetti, L. (2022). Sensitivity studies toward the retrieval of ice crystal habit distributions inside cirrus clouds from upwelling far infrared spectral radiance observations. Journal of Quantitative Spectroscopy and Radiative Transfer, 282, 108120. https://doi.org/10.1016/j.jqsrt.2022.108120

Bi, L., P. Yang, (2017). Improved ice particle optical property simulations in the ultraviolet to far-infrared regime. Journal of Quantitative Spectroscopy and Radiative Transfer, 189, 228-237. https://doi.org/10.1016/j.jqsrt.2016.12.007

Bi, L., P. Yang ( 2014). Accurate simulation of the optical properties of atmospheric ice crystals with invariant imbedding T-matrix method. Journal of Quantitative Spectroscopy and Radiative Transfer, 138,17-35. https://doi.org/10.1016/j.jqsrt.2014.01.013

King MD, Platnick S, Yang P, Arnold GT, Gray MA, Riedi JC, et al. Remote sensing of liquid water and ice cloud optical thickness and effective radius in the arctic: application of airborne multispectral MAS data. J Atmos Oceanic Technol 2004; 21:857–75. doi: https://doi.org/10.1175/1520-0426(2004)021<0857:RSOLWA>2.0.CO;2

Di Natale, G., Palchetti, L., Bianchini, G., and Ridolfi, M.: The two-stream δ-Eddington approximation to simulate the far infrared Earth spectrum for the simultaneous atmospheric and cloud retrieval, Journal of Quantitative Spectroscopy and Radiative Transfer, 246, 106 927, https://doi.org/10.1016/j.jqsrt.2020.106927 , 2020.

---

## Author Comment (AC2)

We thank the reviewer for providing constructive comments. Please find below our answers, along with some summary descriptions of the modifications we made in the revised manuscript we are submitting. For clarity, we also include, **in bold**, the original reviewer's comments.

**Summary**

**This paper presents a sophisticated scheme for modeling ice cloud optical properties by incorporating mixed ice crystal habits, leveraging the upcoming FORUM experiment data. It tests the retrieval scheme on simulated measurements, demonstrating promising performance and identifying the main error components affecting cloud parameter retrievals.**

**Strengths**

- **Innovative Approach:**
  **Develops a new methodology for retrieving ice crystal habit mixtures from spectral radiance measurements, filling a significant gap in atmospheric science.**
- **Utilization of FORUM Data:**
  **The study is forward-looking, leveraging future FORUM experiment measurements to improve ice cloud parameterization, which is crucial for climate modeling.**

We thank the reviewer for the comments and for emphasizing the strengths of the technique we presented in our manuscript.

**Weaknesses**

- **Ill-posed problem:**
  **The concern regarding the manuscript's handling of the ill-posed problem, particularly the issue of solution uniqueness, is a critical aspect that warrants further clarification and enhancement in the manuscript. The ill-posed nature of the problem stems from the inversion process involved in retrieving ice crystal habit fractions from spectral radiance measurements, which, in some scenarios, is shown to be particularly challenging due to the possibility of multiple solutions yielding almost the same minimum of the cost function. I suggest to the authors to expand this point. This will enhance the manuscript strength, providing a more comprehensive understanding of the ill-posed nature of the retrieval problem.**

At the first run of our test retrievals, we found that about 12% of the cases were missing the convergence to the absolute minimum of the cost function (CF). This issue was attributed to the ill-posed nature of our inversion problem. In general, the relatively weak sensitivity of the measurements to the retrieval parameters

decreases the curvature (i.e. makes "flat") the hyper-surface of the CF of which we want to find the absolute minimum. Moreover, the measurement noise and the forward model errors introduce a sort of "roughness" in the CF hyper-surface. Finally, if some parameters included in the state vector are correlated to each other, narrow "canyons" may also characterize the hyper-surface of the CF (see Transtrum et al. 2011 and Ridolfi and Sgheri, 2013). All-together, these elements make the inversion problem ill-posed, and the search of the absolute minimum of the CF becomes a challenging task. The Gauss-Newton method modified with the LM damping, being based on the CF gradient may get trapped on a secondary, local minimum, as actually happened in our first retrieval attempts. A solution could have been to use a stochastic method in place of the LM to find the CF minimum. For example, in Ridolfi and Sheri (2009), the simulated annealing method was used to find the absolute minimum of a CF specifically designed to find the optimal strength of the height-dependent Tikhonov regularization applied to the retrieval of vertical profiles from limb sounding spectral measurements. The simulated annealing, however, as other stochastic minimization methods, requires thousands of evaluations of the CF. In our case, the evaluation of the CF implies the evaluation of the forward model, thus a computationally heavy operation. This feature clearly makes stochastic methods inadequate for our application.

We managed to overcome almost completely this convergence issue by repeating a few times the retrieval starting from slightly different initial guesses of the state vector and by selecting, a posteriori, the solution corresponding to the smallest value of the cost function. With this strategy, only 4 out of the 375 retrievals presented actually miss the absolute minimum of the CF.

It is worth to note that in the simulated retrievals, the cases with a large convergence error are easily detected because in these cases a difference much larger than the error bar exists between the retrieved and the true parameter values. In real data analysis the problem may be harder to identify. A strategy could be to compare the retrieved parameter values and the achieved CF minimum with the respective accumulated statistics, and to treat the outliers as "suspicious" cases. In these cases, restarting the retrieval from different a priori parameter estimates could be beneficial, as it proved to be in the test retrievals presented in the paper. Also a visual inspection of the residuals of the fit for some selected suspicious cases could help to diagnose the problem and to find a workaround.

We included these comments / explanations in Sect. 3.2 of the revised paper.

- **Limited Validation:**
  **The methodology is tested only on simulated data, lacking validation against actual satellite measurements, which raises questions about its real-world applicability.**

Yes, this is a new method that certainly needs to be validated against real measurements, however, we consider such an extensive validation beyond the scope of the current paper. In the near future, we plan to apply the developed retrieval scheme to the large dataset of ground based spectral radiance measurements collected in Antarctica by the Radiation Explored in the Far-InfraRed - Prototype for Applications and Development (REFIR-PAD), a Fourier spectroradiometer deployed by our institute since December 2011 (Palchetti et al., 2015). These measurements are complemented by co-located backscattering/depolarization lidar measurements permitting to estimate the cloud geometrical extension, and by the statistics of ice crystal shapes determined from the measurements of an ice camera deployed on the same site. Potentially, the synergy of these measurements could allow to validate the inversion method presented in this work on the basis of real data and, at the same time, to build a climatology of Antarctic ice cloud crystal habits, corroborated by local in situ measurements.

In the manuscript, we state the need to validate the proposed method against real measurements both in the introduction and at the end of the conclusion section. In the revised version of the manuscript, we further tried to clarify this point.

- **Complexity and Accessibility:**
  **The complex methodology and reliance on specific satellite data may limit its accessibility and applicability by the broader research community.**

  The complexity of the methodology is mainly linked the full-physics model embedded in the proposed method. Simpler methods are easier to use but the information extracted from the measurements is necessarily less exhaustive. In conclusion, we agree that the algorithm proposed may be implemented and exploited only by highly specialized scientists of the field. On the other hand, the method may be applied to the measurements of the forthcoming FORUM and PREFIRE satellite missions, with the potentiality to derive accurate statistics of cloud properties that are extremely relevant for climate studies. In principle, the method could also be applied to the current IASI measurements. However, since IASI measurements are limited to the mid-infrared spectral region, it is not certain whether these measurements contain sufficient information to disentangle the contributions of the various ice crystal shapes to the upwelling spectrum. In the conclusions of the revised manuscript, we added a comment on this possible application of the proposed method.

- **Clarify Methodological Assumptions and Limitations:**
  **The manuscript would benefit from a more detailed discussion of the assumptions underlying the SACR code and the optimal estimation approach used. Addressing the potential limitations these assumptions**

**may pose to the generalizability of the findings will strengthen the manuscript. For instance, how might different atmospheric conditions or cloud compositions affect the retrieval accuracy?**

In the paper, we show that for the mid-latitude and tropical scenarios the algorithm works very well assuming the noise level expected for FORUM measurements. The performance gets worse in polar scenarios, because in these cases clouds are often very close to the ground (on the Antarctic Plateau the ice clouds may be placed even a few hundred meters above the ground) and have a temperature close to that of the ground. In these conditions the cloud emission becomes indistinguishable from that of the ground, thus the inversion becomes strongly ill-conditioned and the probability for the minimization procedure to get trapped into a secondary minimum of the cost function is larger. In these cases the retrieved cloud parameters differ from their true value beyond their error bar. In the revised version of the paper, we discuss how these occurrences could be detected and mitigated in real data analysis (see Sect. 3.2 of the revised paper).

- **Strengthen the Literature Review:**
  **A more thorough review of the current state of research in ice cloud characterization and the retrieval of cloud properties from spectral radiance measurements could provide a stronger foundation for the study. Highlighting the novelty of the approach in the context of existing methodologies will help to underscore the contribution of this work to the field.**

We added the following references in the introduction of the manuscript:

[1] Lolli, S., Campbell, J. R., Lewis, J. R., Gu, Y., and Welton, E. J. , Fu–Liou–Gu and Corti–Peter model performance evaluation for radiative retrievals from cirrus clouds. *Atmospheric Chemistry and Physics*, *17*(11), 7025-7034, (2017).

[2] Campbell, James R., Erica K. Dolinar, Simone Lolli, Gilberto J. Fochesatto, Yu Gu, Jasper R. Lewis, Jared W. Marquis, Theodore M. McHardy, David R. Ryglicki, and Ellsworth J. Welton. "Cirrus cloud top-of-the-atmosphere net daytime forcing in the Alaskan subarctic from ground-based MPLNET monitoring."Journal of Applied Meteorology and Climatology 60, no. 1 (2021): 51-63.

[3] Lewis, J. R., Campbell, J. R., Stewart, S. A., Tan, I., Welton, E. J., and Lolli, S. (2020). Determining cloud thermodynamic phase from the polarized Micro Pulse Lidar. *Atmospheric Measurement Techniques*, *13*(12), 6901-6913.

**Minor Comment**

- **Graphical Representations: Some figures are dense and may be simplified for better clarity and comprehension.**

**Recommendation**

**Given its innovative approach and potential impact on climate modeling, I recommend this paper for acceptance with minor revisions. Addressing the validation with actual satellite data and simplifying complex explanations and figures would significantly enhance the paper's value and readability.**

As mentioned in our answer to the above reviewer's comment titled "Limited validation", in the near future we will extend the validation of our proposed method to cover also the analysis of real measurements. This operation, on its own, is a huge task that we believe is beyond the scope of this paper that, in our view, should be limited to the description of the theoretical basis of the method and on testing its self-consistency and performance on the basis of realistic, synthetic measurements.

The paper we are re-submitting was revised according to the constructive reviewer's comments supplied and, we believe, now should be much easier to read than the original version.

**References**

Ridolfi, M. and Sgheri, L.: A self-adapting and altitude-dependent regularization method for atmospheric profile retrievals, Atmos. Chem. Phys., 9, 1883–1897, https://doi.org/10.5194/acp-9-1883-2009, 2009.

M. Ridolfi and L. Sgheri, "On the choice of retrieval variables in the inversion of remotely sensed atmospheric measurements," Opt. Express **21**, 11465-11474 (2013).

M. K. Transtrum, B. B. Machta, and J. P. Sethna, "Geometry of nonlinear least squares with applications to sloppy models and optimization," Phys. Rev. E 83, 036701 (2011).